# Inhibition of Urban Particulate Matter-Induced Airway Inflammation by RIPK3 through the Regulation of Tight Junction Protein Production

**DOI:** 10.3390/ijms241713320

**Published:** 2023-08-28

**Authors:** Sun-Hee Park, Hyun-Chae Lee, Hye Min Jeong, Jeong-Sang Lee, Hee-Jae Cha, Cheol Hong Kim, Jeongtae Kim, Kyoung Seob Song

**Affiliations:** 1Department of Medical Science, Kosin University College of Medicine, Seo-gu, Busan 49267, Republic of Korea; sunnyday365@naver.com (S.-H.P.); hyun1775@gmail.com (H.-C.L.); todwns@naver.com (H.M.J.); 2Department of Functional Foods and Biotechnology, College of Medical Sciences, Jeonju University, 303 Cheonjam-ro, Jeonju 55069, Republic of Korea; jslee11@jj.ac.kr; 3Department of Genetics, Kosin University College of Medicine, Seo-gu, Busan 49267, Republic of Korea; hcha@kosin.ac.kr; 4Department of Pediatrics, Myongji Hospital, Hanyang University College of Medicine, Goyang 15588, Republic of Korea; chkimmd@hanmail.net; 5Department of Anatomy, Kosin University College of Medicine, Seo-gu, Busan 49267, Republic of Korea; kimjt78@kosin.ac.kr

**Keywords:** urban particulate matter, airway inflammation, RIPK3, tight junction proteins, ROS

## Abstract

Urban particulate matter (UPM) is a high-hazard cause of various diseases in humans, including in the respiratory tract, skin, heart, and even brain. Unfortunately, there is no established treatment for the damage caused by UPM in the respiratory epithelium. In addition, although RIPK3 is known to induce necroptosis, its intracellular role as a negative regulator in human lungs and bronchial epithelia remains unclear. Here, the endogenous expression of RIPK3 was significantly decreased 6 h after exposure to UPM. In RIPK3-ovexpressed cells, RIPK3 was not moved to the cytoplasm from the nucleus. Interestingly, the overexpression of RIPK3 dramatically decreased TEER and F-actin formation. Its overexpression also decreased the expression of genes for pro-inflammatory cytokines (IL-6 and IL-8) and tight junctions (ZO-1, -2, -3, E-cadherin, and claudin) during UPM-induced airway inflammation. Importantly, overexpression of RIPK3 inhibited the UPM-induced ROS production by inhibiting the activation of iNOS and eNOS and by regulating mitochondrial fission processing. In addition, UPM-induced activation of the iκB and NF-κB signaling pathways was dramatically decreased by RIPK3, and the expression of pro-inflammatory cytokines was decreased by inhibiting the iκB signaling pathway. Our data indicated that RIPK3 is essential for the UPM-induced inflammatory microenvironment to maintain homeostasis. Therefore, we suggest that RIPK3 is a potential therapeutic candidate for UPM-induced pulmonary inflammation.

## 1. Introduction

Air pollution is considered to be a critical global risk factor related to developmental activities such as industrialization, urbanization, and agricultural mechanization [1]. Although gas and particulate air pollutants emitted from natural and artificial sources are different, the U.S. Environmental Protection Agency (EPA) has defined six reference air pollutants. These pollutants include particulate matter (PM), sulfur dioxide (SO_2_), carbon monoxide (CO), nitrogen dioxide (NO_2_), ozone (O_3_), and lead (Pb) [1]. Importantly, fine particulate matter (PM_2.5_) can stay in the atmosphere for several hours to days and deeply penetrate into humans’ major organs through the lungs and skin [2,3]. In addition, several studies have suggested that exposure to urban particulate matter (UPM) leads to apoptosis and inflammation of the epithelium of the cornea and airway both in vitro and in vivo [4,5]. Given that UPM causes many diseases in the human body, it is important to understand the physiological mechanisms and the novel therapeutics underlying the negative regulation of UPM-induced airway inflammation.

Receptor-interacting protein kinase-3 (RIPK3, or RIP3) is a critical protein in the programmed/regulated cell death pathway (called necroptosis). Necroptosis is different from other modes of cell death, like apoptosis, because it involves the loss of cell membrane integrity, which triggers the activation of the immune system and a highly proinflammatory response [6]. Shlomovitz et al. suggested that RIPK3 has a dual function in inflammation and cell death [7]. It affects acute and chronic inflammation and plays key roles in sterile and auto-inflammatory pathologies [7]. Recently, Nrf2 deficiency was found to exacerbate PM2.5-induced cardiomyopathy by enhancing oxidative stress, fibrosis, and inflammation through a RIPK3-regulated mitochondrial disorder. These data suggest that RIPK3 induced inflammation by increasing ROS through mitochondrial damage [8]. In addition, Sureshbabu et al. showed that RIPK3 could promote sepsis-triggered acute kidney injury by enhancing mitochondrial dysfunction [9]. Although RIPK3 is a pivotal protein in inflammation and necroptosis, its anti-inflammatory effect during UPM treatment in the airway remains poorly understood.

The tight junction (TJ) system consists of multiple proteins, including transmembrane proteins such as occludin, E-cadherin, claudins, junctional adhesion molecules (JAM), and zona occludens (ZO)-1, -2, and -3 [10]. The association between these proteins and the TJ system is highly dynamic and may play a role in epithelial barrier regulation [10,11]. The epithelial barrier is a protective system that prevents external antigens from infiltrating beyond the barriers of the immune system [12]. Therefore, alteration in the composition and function of TJs might play a critical role in respiratory disease pathogenesis. When epithelial barriers are broken down by external toxins, allergens, or pollutants, pathogens can penetrate the body, which leads to an inflammatory response against them [13]. A breakdown in the epithelial barrier also increases its permeability, which can have critically harmful effects on tissue homeostasis by increasing the body’s exposure to detrimental environmental agents and inducing uncontrollable inflammation [14]. The physiological mechanism by which RIPK3 affects F-actin production and trans-epithelial electrical resistance (TEER) during UPM exposure in the airway epithelial cells remains unclear.

Herein, we characterize whether UPM can affect TJ production to penetrate deep into the cell–cell space. UPM significantly upregulated TJ production. However, RIPK3 overexpression inhibited UPM-induced upregulation of TJ production, proinflammatory cytokine production, F-actin formation, and TEER to control homeostasis. In addition, RIPK3 inhibited UPM-induced inflammation by inhibiting ROS generation by iNOS and eNOS in the cytoplasm. Our data indicate that the RIPK3 protein is essential to control the UPM-induced inflammatory microenvironment by regulating TJ production in the airway epithelium.

## 2. Results

### 2.1. UPM Did Not Affect the Translocation from the Nucleus to the Cytoplasm in BEAS-2B Bronchial Epithelial Cells

To determine whether UPM can control RIPK3 production to regulate UPM-induced inflammation in BEAS-2b cells, we performed qRT-PCR in a time-dependent manner with UPM. RIPK3 expression was dramatically decreased 6 h after treatment with UPM (Figure 1a). In our previous study [14], RIPK3 expression was observed mainly in the nuclear region without LPS treatment, whereas it strongly translocated to the cytoplasmic region after LPS treatment, but not siRNA-RIPK3. To determine the location of RIPK3 expression after UPM treatment, we performed immunocytochemistry. Interestingly, RIPK3 expression was mainly detected in the nuclear and cytoplasmic regions with UPM treatment. However, after UPM treatment in cells transfected with WT RIPK3 or siRNA-RIPK3, the translocation of RIPK3 from the nucleus to the cytoplasm did not occur (Figure 1b). IgG staining was used as a negative control.

### 2.2. RIPK3 Overproduction Downregulates the UPM-Induced Gene Expression of Pro-Inflammatory Cytokines and Tight Junction Proteins

We hypothesized that RIPK3 overexpression may affect UPM-induced airway inflammation to maintain homeostasis. To prove this hypothesis, either RIPK3 overexpression plasmid or siRNA-RIPK3 was transfected into the cells. RIPK3 overexpression dramatically inhibited the UPM-induced gene expression of *pro-inflammatory cytokines* (IL-6 and IL-8) after treatment of UPM at 2 h. However, siRNA-RIPK3 restored the gene expression of *pro-inflammatory cytokines* (Figure 2a). siRNA-control was used as a negative control. In addition, we checked the production of TJ proteins. As expected, RIPK3 overexpression significantly decreased the production of ZO-1, ZO-2, ZO-3, E-cadherin, and claudin after treatment of UPM at 2 h, but not siRNA-RIPK3 (Figure 2b). These findings suggest that RIPK3 overexpression can control the inflammatory microenvironment by regulating TJ production after UPM treatment in human bronchial epithelial cells.

### 2.3. Overexpression of RIPK3 Regulates UPM-Induced TEER and F-Actin Production in Cells

Trans-epithelial electrical resistance (TEER) is a quantitative technique widely used to measure the integrity of TJ dynamics in cell culture models of epithelial monolayers [15]. TEER values are strong indicators of the integrity of the cellular barriers before they are evaluated for drug or chemical transport [15]. According to Figure 2, since the production of TJs was decreased by RIPK3 overexpression, TEER was performed to determine whether the latter can affect membrane integrity during UPM-induced airway inflammation. We found that TEER was reinstated by RIPK3 overexpression, but not by siRNA-RIPK3 (Figure 3a). More interestingly, F-actin was robustly generated by UPM. In the cells exposed to UPM and transfected with WT RIPK3, F-actin production, apical extension, and protrusion were all decreased compared to cells exposed to UPM only (Figure 3b). However, there were no such changes in the cells transfected with siRNA-RIPK3. These results suggest that overexpression of RIPK3 can regulate the inflammatory microenvironment to maintain homeostasis after UPM treatment in human bronchial epithelial cells.

### 2.4. Overexpression of RIPK3 Decreases UPM-Induced ROS Production by Inhibiting Production of eNOS and iNOS

To determine which signal pathway is activated within the cells stimulated by UPM, we evaluated NOS proteins using total and phospho-specific antibodies (Figure 4a). The activation of eNOS by UPM significantly increased at 1 h and decreased at 4 h after UPM stimulation. The phosphorylation of eNOS (Ser1177) gradually increased from 1 h and peaked significantly at 6 h. Interestingly, activation of iNOS significantly increased from 2 h to a maximum of 6 h. We examined whether RIPK3 affects UPM-induced production of eNOS and iNOS. RIPK3 overexpression inhibited the production of eNOS (treatment for 2 h) and iNOS (treatment for 6 h). Interestingly, the phosphorylation of eNOS (Ser1177; treatment for 6 h) was also inhibited by RIPK3 overexpression, but not by siRNA-RIPK3 (Figure 4b). In addition, RIPK3 overexpression could inhibit UPM-induced ROS production, but not siRNA-RIPK3 (Figure 4c). These results suggest that RIPK3 overexpression robustly decreases UPM-induced ROS production by inhibiting the activation of eNOS and iNOS.

### 2.5. RIPK3 Overexpression Upregulates UPM-Induced Mitochondrial Membrane Integrity by Regulating Mitochondrial Fission

To determine which RIPK3 affects UPM-induced mitochondrial membrane integrity, Biotracker 488 mitochondrial dye was used. RIPK3 overexpression significantly increased the staining compared to UPM only (Figure 5a). These findings suggest that UPM induced mitochondrial damage by ROS production, whereas RIPK3 restored the UPM-induced mitochondrial membrane integrity. However, siRNA-RIPK3 did not have an effect. Interestingly, Drp1 expression significantly increased after UPM exposure for 30 min, and then decreased when using the monoclonal Drp1 antibody (Figure 5b). However, RIPK3 overexpression significantly decreased Drp1 expression compared to UPM alone (Figure 5c), suggesting that RIPK3 can regulate mitochondrial fission by controlling Drp1 expression to decrease UPM-induced airway inflammation.

### 2.6. Overexpression of RIPK3 Diminishes UPM-Induced Pro-Inflammatory Cytokine Production by Inhibiting ikB Activation

Mixed lineage kinase domain-like (MLKL) plays a role downstream of RIPK3 in the necroptosis signaling pathway [16,17]. To examine whether RIPK3 affects *MLKL* gene expression in UPM-treated BEAS-2b cells, qPCR of the *MLKL* gene was performed (Figure 6a). *MLKL* gene expression was slightly increased by UPM treatment, but neither overexpression RIPK3 nor siRNA-RIPK3 could affect *MLKL* gene expression in the cells, suggesting that RIPK3 may be not related to MLKL-mediated necroptosis, at least in part, in our system. Next, to examine which signal transduction pathway is induced in the cells stimulated by UPM, UPM treatment was applied in a time-dependent manner. Western blot analysis was performed using a phospho-specific iκB antibody (Figure 6b). UPM-induced phosphorylation of iκB peaked significantly at 45 min and then decreased. Interestingly, overexpression of RIPK3 significantly inhibited the phosphorylation of both iκB and NF-κB, but not siRNA-RIPK3 (Figure 6c). In addition, to investigate whether RIPK3 can affect the NF-κB pathway to regulate UPM-induced production of pro-inflammatory cytokines, p65 overexpression construct was used as a downstream signaling protein of iκB. Overexpression of p65 robustly activated UPM-induced production of IL-6 and IL-8. Interestingly, cotransfection with both RIPK3 and p65 constructs dramatically abolished their expression (Figure 6d), while cotransfection with both siRNA-RIPK3 and wild-type p65 constructs did not. These results suggest that RIPK3 could inhibit UPM-induced production of pro-inflammatory cytokines by blocking the iκB and NF-κB pathways.

## 3. Discussion

There is worldwide concern regarding the serious problems that UPM can cause. Although the worst problem associated with UPM is its effect on human health, there are no well-established treatments for it. In addition, the severity of UPM continues to intensify due to industrialization, civilization, and urbanization. UPM causes many problems not only in the respiratory tract but also in other tissues. Many scientists are trying to remove or diminish the severity of its impact. However, only a few signaling pathway inhibitors have been found to decrease the incidence/severity of UPM-induced respiratory diseases. Herein, we suggest that RIPK3 overexpression dramatically reduces UPM-induced airway inflammation by inhibiting ROS production and acts as a negative regulator protein in normal human bronchial epithelial cells.

RIPK3 is known to induce necroptosis, which is a highly inflammatory process [6,18,19]. Apoptosis, by contrast, is recognized as having different physiological results from necroptosis and is thought to mainly occur without causing inflammation [20,21]. In our previous study [14], RIPK3 significantly inhibited *P. aeruginosa* LPS-induced airway inflammation by regulating post-translational modification and localization. This is a worthy investigation of the anti-inflammatory function of RIPK3 as a negative regulator of inflammation, except for necroptosis. Recently, Ge et al. reported that RIPK3 expression was increased by PM2.5 in *Nrf^-/-^* mice, resulting in increased PM2.5-induced cardiovascular disease [8]. However, RIPK3 overexpression dramatically inhibited UPM-induced airway inflammation in our system (Figure 1). The differences between our results and the former study can be explained as follows: (1) Since the content of F-actin varies between different cells, anti-inflammation caused by F-actin may appear different from one cell to another. In addition, although the amounts of F-actin and TJ expression at intercellular junctions in BEAS-2b cells are relatively small compared to those in other cells (NCI-H292, A549, primary bronchial epithelial cells, and 16HBE14o cells) [22], there are negative regulators such as RIPK3 that are essential for anti-inflammatory effects using the dynamics of F-actin and TJs (Figure 2b and Figure 3). (2) Intracellular physiological functions may change depending on the level of expression and type of posttranslational modification. Recently, Moriwaki et al. reported that RIPK3 activation was tightly regulated by phosphorylation, ubiquitination, and caspase-mediated cleavage [23,24]. Altogether, these post-translational modifications regulate the assembly of a macromolecular signaling complex termed the necrosome [23,24]. In our system, RIPK3 is translocated from the nucleus to the cytoplasm by histone methylation, after which it can become a negative regulator of *P. aeruginosa* LPS-induced airway inflammation [14]. In addition, RIPK3 expression was significantly decreased in pathogenic human lungs compared to that in normal lungs [14]. For these reasons, its function seems to be influenced by the level of RIPK3 expression and its own post-translational modifications. (3) A specific protein may interact with intracellular RIPK3 to regulate the specificity that heavily depends on the inflamed microenvironment. Choi et al. identified Pepellino E3 ubiquitin protein ligase1 (PELI1), which is an E3 ligase that targets RIPK3 for proteasome-dependent degradation [25]. In addition, FK506-binding protein 12 (FKBP12, also known as FKBP1A) was essential to initiating necrosome formation and activating the RIPK1–RIPK3–MLKL signaling pathway in response to TNF receptor 1 ligation [26]. These results strongly suggest that the function of RIPK3 can vary and is dependent on the cell type and binding partners.

The tight junction consists of multiple proteins. The intracellular portions of these transmembrane proteins interact with cytoplasmic peripheral membrane proteins, including ZO-1, -2, -3, and cingulin [27]. Inflammatory bowel disease (IBD) patients have a depletion of TJ barrier function, increased pro-inflammatory cytokine production, and immune dysregulation; however, the relationship between these events is incompletely understood [10]. The disruption of the TJ proteins increases permeability, making it easier for external pathogens to penetrate. Infiltrating pathogens can damage the basolateral membrane in the cell–cell interaction space and infiltrate the blood. Therefore, damage to the TJ proteins can cause serious inflammation in the airway. RIPK3 tended to increase the expression of TJ proteins initially, followed by a gradual decrease. This phenomenon is thought to be due to the fact that RIPK3 reduces inflammation, thereby making TJ protein expression unnecessary (Figure 2b). However, there is a need for further research to clarify the physiological mechanisms by which RIPK3 decreases the expression of TJ proteins. In this study, we present new correlations of the RIPK3–TJ–inflammation relationship, which provides important evidence useful for reducing respiratory inflammation caused by UPM.

UPM is a known inducer of ROS in nasal cells [28]. Upregulation of RIPK3 promotes necroptosis via a ROS-dependent NF-κB pathway, which induces chronic inflammation in HK-2 cells [29]. RIPK3 targets the pyruvate dehydrogenase complex to increase aerobic respiration in TNF-induced necroptosis [30]. However, in our system, RIPK3 significantly inhibited UPM-induced ROS production via the inhibition of eNOS and iNOS activations (Figure 4c). This is a noteworthy finding because RIPK3-dependent reduction in ROS production diminished the pathologies related to oxidative stress. Interestingly, Feng et al. reported that RIPK3 translocation into mitochondria increased Mitofilin degradation to upregulate inflammation and kidney injury after renal ischemia–reperfusion [31]. In addition, RIPK3 induced mitochondrial apoptosis in cardiac IR injury [32]. This discrepancy also seems to be due to the degree of RIPK3 expression and cell specificity. However, more investigation is needed because this difference is a completely dissimilar result from those of other studies.

Recently, several scientists reported that RIPK3 played various roles in mitochondria. RIPK3 induced mitochondrial apoptosis by inhibiting mitophagy in cardiac injury [32], RIPK3 translocated into mitochondria increased inflammation and kidney injury [31], and RIPK3 decreased mitochondrial bioenergetics in metabolic liver diseases [33]. Interestingly, however, RIPK3 can overcome inflammation by inhibiting UPM-induced mitochondrial fission in our system (Figure 5). As mentioned above, we think that this difference is due to the difference between cell specificity and the microenvironment of acute respiratory inflammation. All of the above studies [31,32,33] focused on phenomena that occur in chronic diseases, and there are definite differences between these and the prevalence of acute inflammation such as UPM.

RIPK3 is involved in the regulation of cell death and inflammation. Under certain conditions, RIPK3 could induce inflammation through MLKL-dependent or independent pathways. In the MLKL-independent pathway, RIPK3 activated by stimulants could interact with other protein components of the cell signaling machinery, such as kinases and adaptor proteins. These interactions could lead to the activation of various inflammatory pathways, including the NF-κB or MAPK pathways [6]. Herein, RIPK3 overexpression did not affect *MLKL* gene expression, whereas it dramatically inhibited iκB activation to alter p65-induced pro-inflammatory cytokine production (Figure 6), suggesting that intracellular signaling by UPM was transferred MLKL-independent and ikB-dependent signaling, which was blocked by RIPK3 to suppress the airway inflammation. Because the crosstalk between RIPK3 and NF-κB pathways through MLKL-independent signaling is not well-documented, investigation of anti-inflammatory intracellular signaling by RIPK3 seems to be worthwhile.

We found that RIPK3 expression had an inhibitory effect on UPM-induced airway inflammation and TEER. In addition, RIPK3 overexpression decreased the activation of the ikB and NF-kB signaling pathways and inhibited ROS production by regulating eNOS and iNOS activation. RIPK3 overexpression also inhibited UPM-induced mitochondrial membrane integrity by regulating mitochondrial fission (Figure 6e). Thus, these results suggest that RIPK3 may be a therapeutic candidate in UPM-induced respiratory diseases.

## 4. Materials and Methods

### 4.1. Materials and Cell Culture

The urban particulate matter (UPM) was purchased from Merck. Human bronchial epithelial (BEAS-2b) cells were obtained from ATCC (CRL-9609) and were incubated in BEBM (Lonza, Basel, Switzerland) with a BEGM kit at 37 °C in a humidified incubator with 5% CO_2_. The culture plates were pre-coated with a mixture of 0.01 mg/mL fibronectin (Sigma-Aldrich, St. Louis, MO, USA), 2 mg/mL gelatin (Sigma-Aldrich, St. Louis, MO, USA), and 0.01 mg/mL bovine serum albumin (Gibco BRL, Thermo Fisher; Waltham, MA, USA) dissolved in BEBM medium. The plasmids expressing wild-type RIPK3 construct and wild-type p65 construct were kindly gifted by Dr. You-Sun Kim (Ajou Univ School of Medicine, Suwon, Republic of Korea) [25] and by Dr. Kwang Chul Chung (Yonsei University, Seoul, Republic of Korea), respectively. siRNA-RIPK3 (CUCUCCGCGAAAGGACCAA (dTdT)), siRNA-p65 (GAUGAGAUCUUCCUACUGU(dTdT)), and siRNA-control (CCUACGCCACCAAUUUCGU(dTdT)) were synthesized by Bioneer (Daejeon, Republic of Korea).

### 4.2. Real-Time qRT-PCR

Total RNA was isolated using TRIzol (Thermofisher, St. Louis, MO, USA) from BEAS-2B cells treated with UPM (25 mg/mL). cDNA was synthesized by AccuPower CycleScript RT premix (dT20; Bioneer). Real-time PCR was performed using a QuantStudio 3 Real-Time PCR system (Thermofisher, St. Louis, MO, USA) with TOPreal^TM^ SYBR Green qPCR MIX (enzynomics, Takara, Tokyo, Japan) (a). All reactions were carried out in a total volume of 20 µL which included 10 µL of 2× SYBR Green PCR Master Mix, 300 nM of each primer, and 1 µg of cDNA template. Real-time RT-PCR was performed on a MiniOption Real-Time PCR Detection System (Bio-Rad, Upper Coomera, Australia). The parameters were 95 °C for 10 min, followed by 30~40 cycles of 95 °C for 15 s, 60 °C for 30 s, and 72 °C for 30 s [34]. All reactions were performed in more than triplicate. The relative amount of mRNA was obtained using the comparative cycle threshold method and was normalized using β2-microglobulin as a loading control. The detailed information on the primers we used is summarized in Table 1.

### 4.3. Immunocytochemistry

The cells were fixed with 4% paraformaldehyde, permeabilized with 0.5% Triton X-100 for 5 min, blocked with 5% BSA for 1 h, incubated with diluted (1:100) anti-RIPK3 antibody, and incubated at 4 °C overnight. The secondary antibody, Alexa Fluor 546, was diluted 1:100 in PBS containing 5% BSA and incubated for 1 h at RT. The nuclei were stained with diluted Deep Red (1:300).

### 4.4. Trans-Epithelial Electrical Resistance (TEER)

Before analysis, the electrodes were sterilized and assessed according to the manufacturer’s instructions (Merck, Sigma-Aldrich, St. Louis, MO, USA). The shorter tip was set in the culture plate insert and the longer tip was set in the outer well. The unit area resistance (Ω × cm^2^) was determined by multiplying the sample resistance (Ω) by the effective area of the membrane (4.2 cm^2^ for 6-well Millicell inserts).

### 4.5. F-Actin Staining

F-actin staining was performed using ActinRed 555 ReadyProbe reagent (molecular probes) following the manufacturer’s instructions [14]. The Images were obtained using a Nikon Eclipse 80i microscope (Eclipse 80i, Nikon, Tokyo, Japan) with a 488 nm excitation filter and a 532 nm emission filter.

### 4.6. Western Blot Analysis

We purchased antibodies against phospho-eNOS (Ser1177, #9571), eNOS (#9586), phospho-IκBα (Ser32, #2859), IκBα (#9242), phospho-NF-κB p65 (Ser536, #3033), NF-κB p65 (#8242), and β-actin (#8457) from Cell Signaling Technology; iNOS (ab79342) from Abcam; Drp1 (sc-271583) from Santa Cruz. Anti-mouse IgG HRP (#7076) and anti-rabbit IgG (#7074) HRP were purchased from Cell Signaling Technology as secondary antibodies. The cells were lysed with RIPA lysis buffer (50 mM Tris pH 7.4, 250 mM NaCl, 1% NP40, 0.05% SDS, 2 mM EDTA, 0.5% Deoxycholic acid, 10 mM β-glycerol phosphate, 5 mM NaF, 1 mM Na3VO4, and protease inhibitor cocktail). Equal amounts of whole cell lysates were resolved by SDS-PAGE and transferred to the PVDF membrane. The membranes were blocked with 5% BSA for 1 h at room temperature. Blots were then incubated overnight with a specific antibody in TTBS (0.5% Tween 20 in Tris-buffered saline), respectively. After rinsing with TTBS, the blots were further incubated for 45 min at room temperature with a secondary antibody in TTBS and visualized using the ECL system.

### 4.7. ROS Measurement

The level of intracellular ROS was measured using the DCF-DA method. In brief, BEAS-2b cells were cultured in BEGM media so that on the day of the experiment, there were at least 1~2 × 10^5^ cells per assay. The cells were treated with UPM at a concentration of 25 μg/mL in a time-dependent manner. Next, the cells were trypsinized and dispersed in a single-cell suspension by gently pipetting them up and down. The cells were then stained with the DCF-DA solution (final 10 μM) for 30 min at 37 °C. The intensity of DCF-DA fluorescence was determined using a CytoFLEX (BECKMAN COULTER, Brea, CA, USA) with an excitation wavelength of 488 nm and an emission wavelength of 535 nm.

### 4.8. Analyses and Measurements of Mitochondrial Morphology

BEAS-2b cells were cultured on a coverslip inside a culture dish filled with BEGM medium. When the cells reached the desired confluency, the medium was removed and 500 nM of MitoTracker^®^ Green FM (Thermo Fisher Scientific, St. Louis, MO, USA) was added to visualize the mitochondrial network. The mixture was incubated for 15~45 min at 37 °C. After the staining was completed, the staining solution was replaced with fresh media, and the mitochondrial morphology was observed using fluorescence microscopy.

### 4.9. Statistical Analysis

The data are presented as means ± standard deviations of more than three independent experiments. When appropriate, the statistical differences were measured using Wilcoxon Mann–Whitney tests. A *p*-value < 0.05 was considered statistically significant.

## Figures and Tables

**Figure 1 ijms-24-13320-f001:**
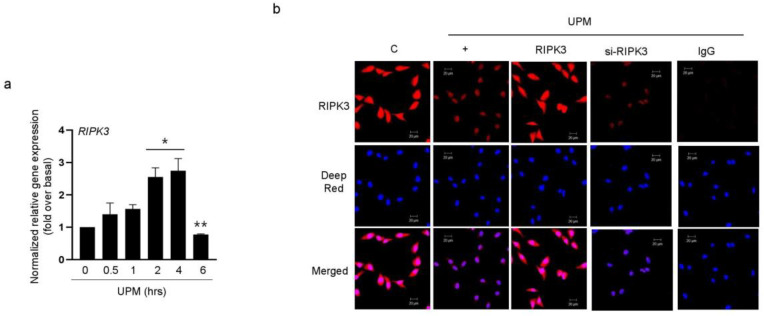
RIPK3 expression was decreased after treatment with UPM in BEAS-2b cells. (**a**) BEAS-2b cells were treated with UPM (25 µg/mL) in a time-dependent manner (0~6 h). The cell lysates were assessed by qRT-PCR. * *p* < 0.05 compared to the control; ** *p* < 0.05 compared to UPM treatment for 4 h. (**b**) The cells were incubated on a chamber slide, fixed with 4% paraformaldehyde, processed for immunofluorescence with RIPK3-specific antibody, and incubated with Alexa Fluor 546 antibody (red). The nuclei were stained with Deep Red. IgG antibody was used as a negative control. “C” stands for control. The scale bar represents 20 μm. All data shown are representative of three independent experiments.

**Figure 2 ijms-24-13320-f002:**
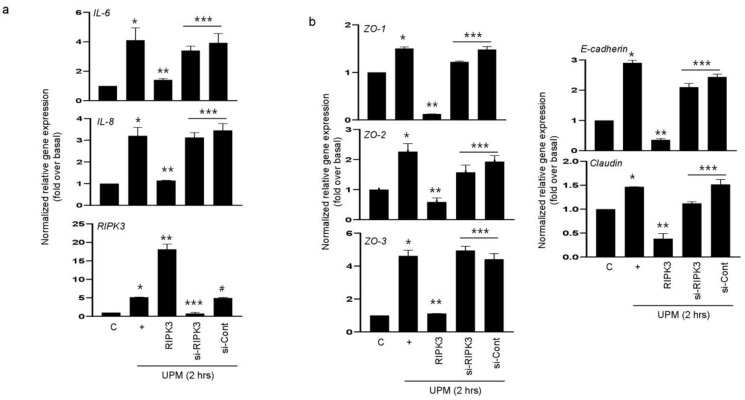
Overexpression of RIPK3 dramatically decreased UPM-induced *pro-inflammatory cytokines* and *TJ* gene expression in BEAS-2b cells. BEAS-2b cells were transfected with either wild-type RIPK3 or siRNA-RIPK3 construct and were then incubated with UPM for 2 h before processing for real-time qRT-PCR. The pro-inflammatory cytokine (**a**) and TJ (**b**) transcripts were analyzed by qRT-PCR. * *p* < 0.05 compared to the control, ** *p* < 0.05 compared to UPM only, *** *p* < 0.05 compared to RIPK3-transfected cells, and ^#^
*p* < 0.05 compared to siRNA-RIPK3-transfected cells. All data shown are representative of three independent experiments.

**Figure 3 ijms-24-13320-f003:**
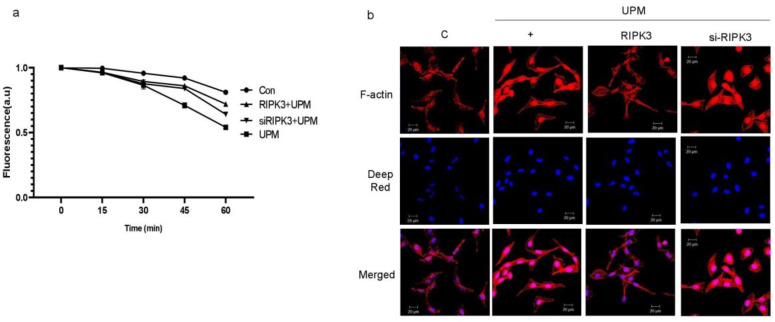
UPM-induced TEER and F-actin production were significantly decreased in RIPK3-overexpressed BEAS-2b cells. (**a**) After transfection, the cells were treated with UPM in a time-dependent manner. TEER testing was performed. All of the data shown are representative of three independent experiments. (**b**) Cells were transfected with either wild-type RIPK3 construct or RIPK3-specific siRNA. Cells were then treated with UPM for 2 h. F-actin staining was performed using ActinRed 555 ReadyProbe reagent (Molecular Probes) following the manufacturer’s instructions. Cell nuclei were stained with diluted Deep Red (1:300). All the data shown are representative of three independent experiments.

**Figure 4 ijms-24-13320-f004:**
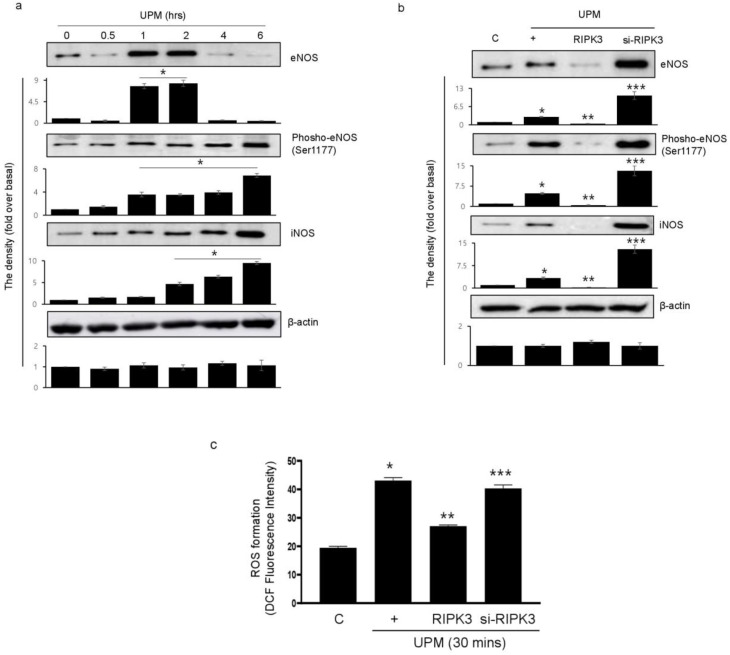
UPM-induced activation of eNOS and iNOS is robustly diminished by overexpression of RIPK3 in BEAS-2b cells. (**a**) The cells were treated with UPM in a time-dependent manner (0~6 h). The total and phospho-eNOS or iNOS expressions were assessed using Western blot analysis with specific antibodies. β-actin was used as an internal loading control. Inserted as a graph below each blot are the densitometer results. * *p* < 0.05 compared to the control. (**b**) The cells were transfected with either wild-type RIPK3 construct or siRNA-RIPK3 and were then treated with UPM for 2 h (eNOS) or 6 h (phospho-eNOS and iNOS) before the generation of total cell lysates. * *p* < 0.05 compared to the control, ** *p* < 0.05 compared to UPM only, and *** *p* < 0.05 compared to RIPK3-transfected cells. (**c**) The cells were transfected with WT RIPK3 and siRNA-RIPK3, and were then treated with the UPM for 30 min. The levels of intracellular ROS were assessed using flow cytometry after DCF-DA staining. All of the data shown are representative of three independent experiments. * *p* < 0.05 compared to the control, ** *p* < 0.05 compared to UPM only, and *** *p* < 0.05 compared to RIPK3-transfected cells.

**Figure 5 ijms-24-13320-f005:**
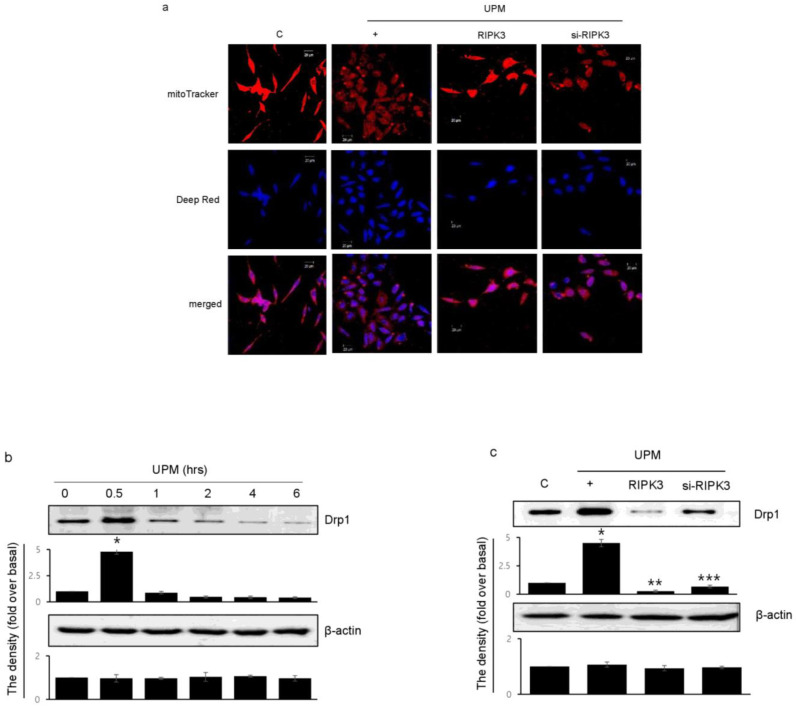
Effects of RIPK3 overexpression on UPM-induced mitochondrial fission in BEAS-2b cells. (**a**) After transfection with WT RIPK3 and siRNA-RIPK3, the cells were treated with UPM for 30 min. The mitochondrial morphology was stained using MitoTracker^®^ Green FM and visualized. (**b**) The cells were treated with UPM in a time-dependent manner (0~6 h). Drp1 expression was analyzed using Western blot analysis with a Drp1-specific antibody. β-actin was used as an internal loading control. * *p* < 0.05 compared to the control. (**c**) The cells were transfected with either wild-type RIPK3 or siRNA-RIPK3 construct and were then incubated with UPM for 30 min before the generation of total cell lysates. The activation of Drp1 was then analyzed by Western blot analysis. * *p* < 0.05 compared to the control, ** *p* < 0.05 compared to UPM only, and *** *p* < 0.05 compared to RIPK3-transfected cells. All of the data shown are representative of three independent experiments.

**Figure 6 ijms-24-13320-f006:**
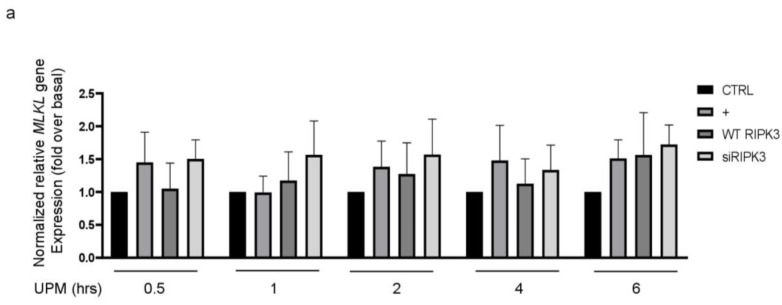
Overexpressed RIPK3 hampers UPM-induced production of pro-inflammatory cytokines by inhibiting the iκB pathway in BEAS-2b cells. (**a**) The cells were transfected with either wild-type RIPK3 or siRNA-RIPK3 construct and were then treated with UPM in a time-dependent manner (0.5 ~ 6 h) prior to generating the total cell lysates for the qPCR of the *MLKL* gene. (**b**) The cells were incubated with UPM (25 μg/mL) in a time-dependent manner (30~60 min). The iκB activation was assessed by Western blot analysis. * *p* < 0.05 compared to the control. (**c**) The cells were transfected with either wild-type RIPK3 or siRNA-RIPK3 construct and were then treated with UPM for 45 min prior to generating the total cell lysates for Western blotting. * *p* < 0.05 compared to the control, ** *p* < 0.05 compared to UPM only, and *** *p* < 0.05 compared to RIPK3-transfected cells. (**d**) The cells were transiently transfected with the wild-type p65 construct and either wild-type or siRNA constructs of RIPK3, and cells were incubated with UPM for 2 h. The lysates were prepared for qPCR. * *p* < 0.05 compared with control; ** *p* < 0.05 compared with UPM treatment; *** *p* < 0.05 compared with wild-type RIPK3 transfectants; ^#^
*p* < 0.05 compared with wild-type p65 transfectants; ^##^
*p* < 0.05 compared with wild-type RIPK3 and wild-type p65 transfectants. All the data shown are representative of three independent experiments. (**e**) A schematic model illustrating that RIPK3 overexpression inhibits UPM-induced airway inflammation by regulating the activation of iκB and NF-κB signaling pathways and ROS production.

**Table 1 ijms-24-13320-t001:** The information on primers for qRT-PCR.

Gene	Sequence
GAPDH	F: CCACATGGCCTCCAAGGAGTAAGAC
	R: AGGAGGGGAGATTCAGTGTGGTGGG
IL-6	F: AGACAGCCACTCACCTCTTCAG
	R: TTCTGCCAGTGCCTCTTTGCTG
IL-8	F: GAGAGTGATTGAGAGTGGACCAC
	R: CACAACCCTCTGCACCCAGTTT
RIPK3	F: GCTACGATGTGGCGGTCAAGA
	R: TTGGCCCAGTTCACCTCTCG
ZO-1	F: GTCCAGAATCTCGGAAAAGTGCC
	R: CTTTCAGCGCACCATACCAACC
ZO-2	F: ATTAGTGCGGGAGGATGCCGTT
	R: TCTGCCACAAGCCAGGATGTCT
ZO-3	F: GCTTCCTCAAGGGCAAGAGCAT
	R: CGTGTCAGGTTCTGGAATGGCA
E-cadherin	F: GTCTGTAGGAAGGCACAGCC
	R: TGCAACGTCGTTACGAGTCA
Claudin	F: CAGCTGTTGGGCTTCATTCT
	R: ATCACTCCCAGGAGGATGCC
MLKL	F: TCACACTTGGCAAGCGCATGGT
	R: GTAGCCTTGAGTTACCAGGAAGT

## Data Availability

The datasets used and/or analyzed during the current study are available from the corresponding author upon reasonable request.

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
