# Peer review of "Inhibition of Urban Particulate Matter-Induced Airway Inflammation by RIPK3 through the Regulation of Tight Junction Protein Production"

_ijms, 2023, doi:10.3390/ijms241713320_

Round 1

Reviewer 1 Report (Previous Reviewer 1)

My previous review included :

1.     All western blot figures (Fig 4a, b, Fig 5b,c and Fig 6b,c) should include a densitometry plot with relevant statistics.

In the current manuscript there were no Figures.

Please upload again with the revised figures 

Author Response

Reviewer 2 Report (Previous Reviewer 3)

The authors have addressed my concerns. I have no further comment.

Round 2

Reviewer 1 Report (Previous Reviewer 1)

Since I have not seen the figures in the manuscript I wanted to re -confirm, that the figures are presented in the manuscript.

Author Response

We deeply appreciate it. Please upload again with the revised figures as an attachment.

Round 3

Reviewer 1 Report (Previous Reviewer 1)

thank you for the update version.

The authors did not include "significant signs" in the modified  plots and did not mention it in the results section. It should be added   

Author Response

This manuscript is a resubmission of an earlier submission. The following is a list of the peer review reports and author responses from that submission.

Round 1

Reviewer 1 Report

The concept of this research is very interesting. Unfortunately parts of the text were copied from their previous paper. Thus, the authors need to revise the paper according to my comments

Major concerns:

1.    Fig 1- paragraph. 2.1 and figure 1 were already published in their previous paper (Yun M. et al. Inhibition of Pseudomonas aeruginosa LPS-Induced airway inflammation by RIPK3 in human airway. J Cell Mol Med. 2022;26(21):5506-5516. ). Authors may cite their previous publication but should not present the same figure/legends/methods. It is not acceptable.

     Moreover in Fig 1, which was also presented in their previous publication, the authors need to describe which cells in the IHC were stained and what is "inflamed human lung"- which pathologic conditions were included?

2.    Methods- the authors copied the same methods from their previous publication (Yun M et al. Inhibition of Pseudomonas aeruginosa LPS-Induced airway inflammation by RIPK3 in human airway. J Cell Mol Med. 2022;26(21):5506-5516) without changing anything, and without citing their previous work.

3.    Methods- a description of western blot and RNA extraction and Real time PCR is missing.

4.    Several papers showed that RIPK3 expression is elevated in lung inflammation (Huang HR, et al. RIPK3 Activates MLKL-mediated Necroptosis and Inflammasome Signaling during Streptococcus Infection. Am J Respir Cell Mol Biol. 2021;64(5):579-591; Lu Z, et al. Necroptosis Signaling Promotes Inflammation, Airway Remodeling, and Emphysema in Chronic Obstructive Pulmonary Disease. Am J Respir Crit Care Med. 2021;204(6):667-681). How do the authors explain these findings compared to their results?

5.    The authors should add the expression of MLKL to their model.

6.    Legends fig 4 /6 – please correct there are some symbols that appeared.

Statistical analysis – "When appropriate, the statistical differences were measured using Wilcoxon Mann-Whitney tests" unclear. What is REF 31? How is it connection to statistical methods? 

Moderate editing of English language required

Author Response

The responses for comments of Reviewer I

  1. Fig 1- paragraph. 2.1 and figure 1 were already published in their previous paper (Yun M. et al. Inhibition of Pseudomonas aeruginosa LPS-Induced airway inflammation by RIPK3 in human airway. J Cell Mol Med. 2022;26(21):5506-5516. ). Authors may cite their previous publication but should not present the same figure/legends/methods. It is not acceptable.

Moreover in Fig 1, which was also presented in their previous publication, the authors need to describe which cells in the IHC were stained and what is "inflamed human lung"- which pathologic conditions were included?

RESPONSE: We decided that Fig.1 (in submitted manuscript) was deleted in revised manuscript as the reviewer’s comment. Our intension was to investigate the effects of RIPK3 in various experiments, but we decided to follow the reviewer’s comments. And BEAS-2B cells, an immortalized but non-tumorigenic epithelial cell line from human bronchial epithelial cells, were transfected with either RIPK3 overexpression construct or siRNA-RIPK3. In this manuscript, we used only BEAS-2B cells.

  1. Methods- the authors copied the same methods from their previous publication (Yun M et al. Inhibition of Pseudomonas aeruginosa LPS-Induced airway inflammation by RIPK3 in human airway. J Cell Mol Med. 2022;26(21):5506-5516) without changing anything, and without citing their previous work.

RESPONSE: We thank for the reviewer’s comments. The methods were changed as much as possible and the citation was added if necessary.

  1. Methods- a description of western blot and RNA extraction and Real time PCR is missing.

RESPONSE: We added the methods of Western blot, RNA extract, and qPCR into the “Materials and Methods” session as follows:

   “4.2 Real-time qRT-PCR

Total RNA was isolated using TRIzol (Thermofisher) from BEAS-2B cells treated with UPM (25 mg/ml). cDNA was synthesized by AccuPower CycleScript RT premix (dT20; Bioneer). Real-time PCR was performed using a QuantStudio 3 Real-Time PCR system(Thermofisher) with TOPrealTM SYBR Green qPCR MIX(enzynomics). All Reactions were carried out in a total volume of 20 µl which included 10 µl of 2x SYBR Green PCR Master Mix, 300 nM of each primer, and 1 µg of cDNA template. Real time RT-PCR was performed on a MiniOption Real-Time PCR Detection System (Bio-Rad). The parameters were 95 °C for 10 min, followed by 30 ~ 40 cycles of 95 °C for 15 sec, 60 °C for 30 sec, and 72 °C for 30 sec [32]. All reactions were performed more than triplicate. The relative amount of mRNA was obtained using the comparative cycle threshold method and was normalized using β2-microglobulin as a loading control.

 4.6 Western blot analysis

The cells were lysed with RIPA lysis buffer (50 mM Tris pH 7.4, 250 mM NaCl, 1% NP40, 0.05% SDS, 2 mM EDTA, 0.5% Deoxycholic acid, 10 mM β-glycerol phosphate, 5 mM NaF, 1 mM Na3VO4, protease inhibitor cocktail). Equal amounts of whole cell lysates were resolved by SDS-PAGE and transferred to the PVDF membrane. The membranes were blocked with 5% BSA for 1 hr at room temperature. Blots were then incubated overnight specific antibody in TTBS (0.5% Tween 20 in Tris-buffered saline), respectively. After rinsing with TTBS, the blots were further incubated for 45 min at room temperature with secondary antibody in TTBS and visualized using the ECL system.

  1. Several papers showed that RIPK3 expression is elevated in lung inflammation (Huang HR, et al. RIPK3 Activates MLKL-mediated Necroptosis and Inflammasome Signaling during Streptococcus Infection. Am J Respir Cell Mol Biol. 2021;64(5):579-591; Lu Z, et al. Necroptosis Signaling Promotes Inflammation, Airway Remodeling, and Emphysema in Chronic Obstructive Pulmonary Disease. Am J Respir Crit Care Med. 2021;204(6):667-681). How do the authors explain these findings compared to their results?

RESPONSE: We presented this similar difference in our previous study (JCMM 2022 26:5506-16). Additionally, two different studies suggested by the reviewer show the differences in cell specificity. They were used the Ripk3-/- mice and macrophages (Am J Respir Cell Mol Biol) and the lung tissue from COPD patients and macrophages (Am J Respir Crit Care Med). However, we used the normal lung tissue (in the submitted manuscript) and an immortalized but non-tumorigenic epithelial cell line from human bronchial epithelial cells (BEAS-2B) to examine the UPM-induced acute inflammation. Due to these differences, RIPK3 may have another new function in normal lung and bronchial epithelial cells. Taken together, these discrepancies are caused by differences in the cell specificity and their own surrounding environments.

  1. The authors should add the expression of MLKL to their model.

RESPONSE: We added the results of UPM-induced MLKL gene expression as a supplement result. The MLKL gene expression was slightly increased by UPM, but the overexpression RIPK3 could not affect MLKL gene expression in the cells transfected with RIPK3, suggesting that RIPK3 may be not related to MLKL-mediated necroptosis, at least in part, in our system.

  1. Legends fig 4 /6 – please correct there are some symbols that appeared.

RESPONSE: We thank for your point. We changed to new ones as follows:

“Fig. 4-(a) After transfection, the cells were treated with UPM in a time-dependent manner. The TEER testing was performed. All of the data shown are representative of three independent experiments. (b) Cells were transfected with either wild-type RIPK3 construct or RIPK3-specific siRNA. Cells were then treated with UPM for 2 hr. F-actin staining was performed using ActinRed 555 ReadyProbe reagent (Molecular Probes) following the manufacturer’s instructions. Their nuclei were stained with diluted Deep-Red (1:300). All the data shown are representative of three independent experiments.”

And

“Fig. 6-(a) After transfection with WT RIPK3 and siRNA-RIPK3, the cells were treated with UPM for 30min.”

  1. Statistical analysis – "When appropriate, the statistical differences were measured using Wilcoxon Mann-Whitney tests" unclear. What is REF 31? How is it connection to statistical methods?

RESPONSE: We deleted the citation of Ref 31.

Reviewer 2 Report

The manuscript “Inhibition of urban particulate matter-induced airway inflammation by RIPK3 through the regulation of tight junction protein production. by Sun-Hee Park et al. investigated the role of PIPK3 in the UPM-induced inflammatory microenvironment. However, there are several concerns with this manuscript:

Major:

1. Please show the efficiency of siRIPK3 by WB.

2. In Figure 5b, its hard to see p-eNOS/iNOS were induced by UPM, please use representative images and do statistics of all WB figures.

3. In Figure 5b, the effect of p-eNOS and iNOS treated with siRNA-RIPK3 was stronger than that induced by UPM. Please comment.

4. The mechanism of how RIPK3 is regulated in the UPM-induced inflammatory microenvironment is not clear.

Minor: 

1. The legends are not clear, what does C,+, RIPK3 mean in all related figures, Is C for control? But the Vehicle was used in images, please confirm.

2. Please check the time of UPM treatment in Figures 4 and 6.

3. Many methods are missing, including western blot, siRNA, qPCR, and so on, please check.

Author Response

The responses for comments of Reviewer II

  1. Please show the efficiency of siRIPK3 by WB.

RESPONSE: We thank for the reviewer’s point. The result of real-time qPCR was added in order to determine the efficiency of either wild-type RIPK3 or siRNA-RIPK3 to Fig. 2a. As seen, RIPK3 expression was significantly increased in the cells transfected with WT RIPK3, whereas it was robustly decreased by siRNA-RIPK3 compared to siRNA-control. The siRNA-control was used as a negative control. For the uniformity of each figure, qPCR results were added to Figure 2a

  1. In Figure 5b, it’s hard to see p-eNOS/iNOS were induced by UPM, please use representative images and do statistics of all WB figures

RESPONSE: We thank for this comment. We changed to new result into Fig. 4B in revised manuscript. In addition, after the intensity was analyzed by the densitometer, we added the quantification in all Western blot analysis as follows:

.

  1. In Figure 5b, the effect of p-eNOS and iNOS treated with siRNA-RIPK3 was stronger than that induced by UPM. Please comment.

RESPONSE: Whenever this experiment was performed, the bands were stronger in the siRNA-RIPK3 group than UPM only. We thought this phenome was because the endogenous RIPK level was completely knocked down by siRNA-RIPK3. The expression level of RIPK3 was added to Fig. 2a (revised manuscript)

  1. The mechanism of how RIPK3 is regulated in the UPM-induced inflammatory microenvironment is not clear.

RESPONSE: This physiological mechanism study was not included in this manuscript because it was difficult to predict because it was different from the previous study (J Cell Mol Med. 2022;26(21):5506-5516). In order to know the physiological mechanism of RIPK3, RNA-Seq analysis was performed to find the inflammation-related gene. In addition, the gain-of function and loss-of-function experiments were conducted to find out whether the identified inflammation-related genes were involved in RIPK3-mediated airway inflammation. Briefly, the mitochondria-related protein, we found, might be essential for RIPK3-mediated protein interaction to regulate the UPM-induced airway inflammation. This study is ongoing and will be submitted to IJMS as soon as possible. We appreciate your understanding.

# Minor

  1. The legends are not clear, what does C,+, RIPK3 mean in all related figures, Is C for control? But the Vehicle was used in images, please confirm.

RESPONSE: “C” stood for the control. And we changed the “Vehicle” to “C” in the images figure.

  1. Please check the time of UPM treatment in Figures 4 and 6.

RESPONSE: We thank for your point. We changed to new ones as follows:

“Fig. 4-(a) After transfection, the cells were treated with UPM in a time-dependent manner. The TEER testing was performed. All of the data shown are representative of three independent experiments. (b) Cells were transfected with either wild-type RIPK3 construct or RIPK3-specific siRNA. Cells were then treated with UPM for 2 hr.

and

“Fig. 6-(a) After transfection with WT RIPK3 and siRNA-RIPK3, the cells were treated with UPM for 30min.”

  1. Many methods are missing, including western blot, siRNA, qPCR, and so on, please check.

RESPONSE: We added the methods of Western blot, RNA extract, and qPCR into the “Materials and Methods” session as follows:

   “4.2 Real-time qRT-PCR

Total RNA was isolated using TRIzol (Thermofisher) from BEAS-2B cells treated with UPM (25 mg/ml). cDNA was synthesized by AccuPower CycleScript RT premix (dT20; Bioneer). Real-time PCR was performed using a QuantStudio 3 Real-Time PCR system(Thermofisher) with TOPrealTM SYBR Green qPCR MIX(enzynomics). All Reactions were carried out in a total volume of 20 µl which included 10 µl of 2x SYBR Green PCR Master Mix, 300 nM of each primer, and 1 µg of cDNA template. Real time RT-PCR was performed on a MiniOption Real-Time PCR Detection System (Bio-Rad). The parameters were 95 °C for 10 min, followed by 30 ~ 40 cycles of 95 °C for 15 sec, 60 °C for 30 sec, and 72 °C for 30 sec [32]. All reactions were performed more than triplicate. The relative amount of mRNA was obtained using the comparative cycle threshold method and was normalized using β2-microglobulin as a loading control.

 4.6 Western blot analysis

The cells were lysed with RIPA lysis buffer (50 mM Tris pH 7.4, 250 mM NaCl, 1% NP40, 0.05% SDS, 2 mM EDTA, 0.5% Deoxycholic acid, 10 mM β-glycerol phosphate, 5 mM NaF, 1 mM Na3VO4, protease inhibitor cocktail). Equal amounts of whole cell lysates were resolved by SDS-PAGE and transferred to the PVDF membrane. The membranes were blocked with 5% BSA for 1 hr at room temperature. Blots were then incubated overnight specific antibody in TTBS (0.5% Tween 20 in Tris-buffered saline), respectively. After rinsing with TTBS, the blots were further incubated for 45 min at room temperature with secondary antibody in TTBS and visualized using the ECL system.

Reviewer 3 Report

The authors should address the following issues: 

1.       In Figure 1, there are multiple instances where the images appear to overlap with those from a previously published paper or the current manuscript. For example, there seems to be an overlap between images #3 and #4 in the inflamed lung tissue group of Figure 1. However, the RIPK3 intensity in these two images differs. Additionally, image #2 in the inflamed lung group is identical to the image labeled Figure 1 (A) #1 in the published manuscript (Yun, M et.al. Inhibition of Pseudomonas aeruginosa LPS-Induced airway inflammation by RIPK3 in human airway. J Cell Mol Med 2022, 26, 5506-5516), which was authored by the same group. Similar overlap was found between #8 in current manuscript and #10 in published manuscript.

2.       In Figure 1, the RIPK3 staining appears to be nonspecific. It would be helpful if the authors could provide images of the IgG control staining for comparison. Furthermore, I am curious to know whether RIPK3 is expressed throughout the entire lung or primarily in the epithelial cells.

3.       The method description lacks certain information, including details about the siRNA and overexpression plasmid utilized in the experiments.

4.       Most of the experimental designs consist with four groups: Vehicle, UMP+, RIPK3 and si-RIPK3. However, I'm curious about the control treatment for si-RIPK3. It seems that the expression of RIPK3 was not decreased in the si-RIPK3 cells when compared to the UMP+ group.

5.       The author should provide further clarification on Figure 2b. The relationship between this particular result and the hypothesis is currently unclear.

6.       The statement that “RIPK3 can regulate mitochondria fission/fusion to maintain homeostasis during UPM-induced airway inflammation” appears to be overstated. As it is solely supported by the observed changes in the Drp1 protein. Further evidence is needed to substantiate this claim.

7.       Quantification of Western Blot is missing.

8.       The statistical analysis of Figure 5c is missing.

Author Response

The responses for comments of Reviewer III

  1. In Figure 1, there are multiple instances where the images appear to overlap with those from a previously published paper or the current manuscript. For example, there seems to be an overlap between images #3 and #4 in the inflamed lung tissue group of Figure 1. However, the RIPK3 intensity in these two images differs. Additionally, image #2 in the inflamed lung group is identical to the image labeled Figure 1 (A) #1 in the published manuscript (Yun, M et.al. Inhibition of Pseudomonas aeruginosa LPS-Induced airway inflammation by RIPK3 in human airway. J Cell Mol Med 2022, 26, 5506-5516), which was authored by the same group. Similar overlap was found between #8 in current manuscript and #10 in published manuscript.

RESPONSE: We decided that Fig.1 (in the submitted manuscript) was deleted in revised manuscript as the reviewer’s comment. Our intension was to investigate the effects of RIPK3 in various experiments, but we decided to follow the reviewer’s comments.

  1. In Figure 1, the RIPK3 staining appears to be nonspecific. It would be helpful if the authors could provide images of the IgG control staining for comparison. Furthermore, I am curious to know whether RIPK3 is expressed throughout the entire lung or primarily in the epithelial cells.

RESPONSE: Because Fig. 1 (in the submitted manuscript) was deleted in this revised manuscript, IgG control staining did not need to perform. However, we also thought that IgG staining as a negative control, is essential to prove the specificity of RIPK3 antibody. Indeed, we added this IgG control staining in Fig. 1B (in the revised version). In addition, we thought that RIPK3 was expressed throughout the entire lung (Fig. 1 in the submitted manuscript), but not epithelial cells. We performed RIPK3 IHC staining many times in BEAS-2B cells. However, RIPK3 did not strongly expressed in BEAS-2B epithelial cells. In addition, RIPK3 expression was decreased in a UPM or LPS time-dependent manner.

  1. The method description lacks certain information, including details about the siRNA and overexpression plasmid utilized in the experiments.

RESPONSE: The sequence of siRNA-RIPK3 was added to “Materials and Methods” session. And the RIPK3 overexpression construct was kindly gifted by Dr. You-Sun Kim (Ajou University School of Medicine, Suwon, Korea; Mol Cell 2018 70:920-35). We added this information about RIPK3 overexpression construct to “Materials and Methods” session as follows:

“The plasmid expressing wild-type RIPK3 construct was kindly gifted by Dr. You-Sun Kim (Ajou Univ School of Medicine, Suwon, Korea)[23] and siRNA-RIPK3 [CUCUCCGCGAAAGGACCAA (dTdT)], and siRNA-control [CCUACGCCACCAAUUUCGU(dTdT)] were synthezied to Bioneer (Daejeon, Korea).”

  1. Most of the experimental designs consist with four groups: Vehicle, UMP+, RIPK3 and si-RIPK3. However, I'm curious about the control treatment for si-RIPK3. It seems that the expression of RIPK3 was not decreased in the si-RIPK3 cells when compared to the UMP+ group.

RESPONSE: We thank for this comment. We changed to new result into Fig. 1B that RIPK3 expression was significantly decreased in the cells transfected with siRNA-RIPK3 compared to vehicle control. siRNA-RIPK3 could robustly decrease the endogenous RIPK3 expression in the cells. We also added the results about siRNA control in entire Fig. 2 (in the revised version). The expression of proinflammatory cytokines and TJs was significantly restored in the cells transfected with siRNA-RIPK3 compared to wild-type RIPK3 transfection (Fig.2 in the revised version). This phenomenon is clearly shown in Fig. 2 (in the revised version), so it is omitted from the next experiments. We appreciate your understanding.

  1. The author should provide further clarification on Figure 2b. The relationship between this particular result and the hypothesis is currently unclear.

RESPONSE: We deeply thank for this point. We made a huge mistake by making several corrections during writing this manuscript. We changed to new interpretation as follows:

 Abstract: in RIPK3-ovexpressed cells, RIPK3 was not moved to the cytoplasm from the nucleus

Results session: However, after UPM treatment in the cells transfected with WT RIPK3 or siRNA-RIPK3, the translocation of RIPK3 from the nucleus to the cytoplasm was not occurred (Figure 1b). IgG staining was used as a negative control.

  1. The statement that “RIPK3 can regulate mitochondria fission/fusion to maintain homeostasis during UPM-induced airway inflammation” appears to be overstated. As it is solely supported by the observed changes in the Drp1 protein. Further evidence is needed to substantiate this claim.

RESPONSE: Your point is definitely right. We overstated the interpretation of Fig. 5 in the revised manuscript. Thus, we changed to new explanation in the Fig. 5 as follows:

“RIPK3 can regulate mitochondria fission by controlling Drp1 expression to decrease UPM

induced airway inflammation.”

  1. Quantification of Western Blot is missing.

RESPONSE: After the intensity was analyzed by the densitometer, we added the quantification in all Western blot analysis.

  1. The statistical analysis of Figure 5c is missing.

RESPONSE: We also thank for your point. We added the statistical analysis in Fig. 4C in the

revised manuscript.

Round 2

Reviewer 1 Report

ijms-2507328

1.     Although the authors responded to most of my comments, I could not understand what Fig.1 is. It seems that it is the same figure from the first version as in the revised version. They changed only the legends in Fig. 1. Unclear.

2.     Moreover as I wrote in my comments ", the authors need to describe which cells in the IHC were stained and what is "inflamed human lung"- which pathologic conditions were included? This information is missing.

3.     My previous comment was:

The authors should add the expression of MLKL to their model.

The authors RESPONSE: We added the results of UPM-induced MLKL gene expression as a supplement result. The MLKL gene expression was slightly increased by UPM, but the overexpression RIPK3 could not affect MLKL gene expression in the cells transfected with RIPK3, suggesting that RIPK3 may be not related to MLKL-mediated necroptosis, at least in part, in our system.

The authors should add this information to the Methods, Results and Discussion. This is important data.

4.     Methods : Real time PCR – add primer sequences; western Blot- add antibody details.

I recommend that the authors read their manuscript carefully and pay attention to details in the Methods, Reference list and legends.

An English language editor should review the manuscript. .

An English language editor should review the manuscript. .

Reviewer 2 Report

Thanks for the authors' response. But for the question 'In Figure 5b, the effect of p-eNOS and iNOS treated with siRNA-RIPK3 was stronger than that induced by UPM. Please comment', the author responded you thought this phenome was because the endogenous RIPK level was completely knocked down by siRNA-RIPK3. If so, what is the role of single siRNA-RIPK3 (without UPM treatment) in each group? 

For the question 'The mechanism of how RIPK3 is regulated in the UPM-induced inflammatory microenvironment is not clear'. the author thinks the mitochondria-related protein might be essential for RIPK3-mediated protein interaction to regulate the UPM-induced airway inflammation. This study is ongoing and will be submitted to IJMS as soon as possible. At this point, the study lacks the kind of mechanism, I'll suggest that the authors resubmit it after doing all those parts.

Reviewer 3 Report

There have been no updates provided regarding any of the figures. It seems that the authors may have been negligent and overlooked this important aspect. Furthermore, the response provided does not adequately address the concerns raised by the reviewers.

Round 3

Reviewer 1 Report

ijms-2507328

The manuscript is well revised according to my comments, nevertheless some details are missing:

1.     All western blot figures (Fig 4a, b, Fig 5b,c and Fig 6b,c) should include a densitometry plot with relevant statistics.

2.     Please add to the methods table 1 – the sequence of MLKL primers

Reviewer 2 Report

1. What fascinates me the most about this article is the human data, unfortunately, it has been published.

2. The mechanism of this study is not clear.

Reviewer 3 Report

A critical issue that undermines the credibility of their conclusion is the absence of comprehensive statistical analysis of all Western blot data. This glaring gap in their research renders the evidence insufficient to substantiate their claims convincingly.